# Open-Set Video-based Facial Expression Recognition with Human Expression-sensitive Prompting

Yuanyuan Liu
School of Computer Science, China
University of Geosciences (Wuhan)
Wuhan, China
liuyy@cug.edu.cn

Yuxuan Huang*
School of Computer Science, China
University of Geosciences (Wuhan)
Wuhan, China
cosinehuang@cug.edu.cn

Shuyang Liu
School of Computer Science, China
University of Geosciences (Wuhan)
Wuhan, China
20171003670@cug.edu.cn

Yibing Zhan*
JD Explore Academy
Beijing, China
zhanyibing@jd.com

Zijing Chen
Cisco - La Trobe Centre for Artificial
Intelligence and Internet of Things,
La Trobe University
Flora Hill, Australia
zijing.chen@latrobe.edu.au

Zhe Chen
Cisco - La Trobe Centre for Artificial
Intelligence and Internet of Things,
La Trobe University
Flora Hill, Australia
zhe.chen@latrobe.edu.au

## Abstract

In Video-based Facial Expression Recognition (V-FER), models are typically trained on closed-set datasets with a fixed number of known classes. However, these models struggle with unknown classes common in real-world scenarios. In this paper, we introduce a challenging Open-set Video-based Facial Expression Recognition (OV-FER) task, aiming to identify both known and new, unseen facial expressions. While existing approaches use large-scale vision-language models like CLIP to identify unseen classes, we argue that these methods may not adequately capture the subtle human expressions needed for OV-FER. To address this limitation, we propose a novel Human Expression-Sensitive Prompting (HESP) mechanism to significantly enhance CLIP's ability to model video-based facial expression details effectively. Our proposed HESP comprises three components: 1) a textual prompting module with learnable prompts to enhance CLIP's textual representation of both known and unknown emotions, 2) a visual prompting module that encodes temporal emotional information from video frames using expression-sensitive attention, equipping CLIP with a new visual modeling ability to extract emotion-rich information, and 3) an open-set multi-task learning scheme that promotes interaction between the textual and visual modules, improving the understanding of novel human emotions in video sequences. Extensive experiments conducted on four OV-FER task settings demonstrate that HESP can significantly boost CLIP's performance (a relative improvement of 17.93% on AUROC and 106.18% on OSCR) and outperform other state-of-the-art open-set video understanding methods by a large margin. Code is available at https://github.com/cosinehuang/HESP.

*Corresponding author

## CCS Concepts

• **Human-centered computing**; • **Computing methodologies → Computer vision**;

## Keywords

Open-Set recognition, video-based facial expression recognition, textual prompting, visual prompting, CLIP

**ACM Reference Format:**
Yuanyuan Liu, Yuxuan Huang, Shuyang Liu, Yibing Zhan, Zijing Chen, and Zhe Chen. 2024. Open-Set Video-based Facial Expression Recognition with Human Expression-sensitive Prompting. In *Proceedings of the 32nd ACM International Conference on Multimedia (MM '24), October 28-November 1, 2024, Melbourne, VIC, Australia.* ACM, New York, NY, USA, 10 pages. https://doi.org/10.1145/3664647.3681583

## 1 Introduction

Video-based Facial Expression Recognition (V-FER) targets the recognition of predetermined emotion categories from video data. Despite its advancements in enhancing human-centered visual comprehension, this method usually focuses on a limited set of predefined categories. This restriction does not capture the full complexity and diversity of human expressions, limiting its ability to recognize a broader range of emotional states. To address this limitation, we introduce the Open-set Video-based Facial Expression Recognition (OV-FER) task, which seeks to identify known and discover new human emotions from videos. Similar to other open-set paradigms [4, 7, 20, 27, 28], the OV-FER requires the ability to recognize and classify previously unseen or 'unknown' expressions. Rooted in human emotion understanding, an effective OV-FER model is supposed to provide a robust understanding of various facial expression patterns in diverse and unpredictable environments, thus benefiting diversified applications like smart healthcare [5], human-computer interaction [6], and driver assistance system [35].

Recently, several studies [21, 30, 34] have proposed leveraging large-scale pre-trained models like CLIP [29] to facilitate open-set recognition tasks. With CLIP's remarkable generalization capabilities, it becomes feasible to effectively identify new categories from open-set data. However, when applying CLIP to video analysis, a

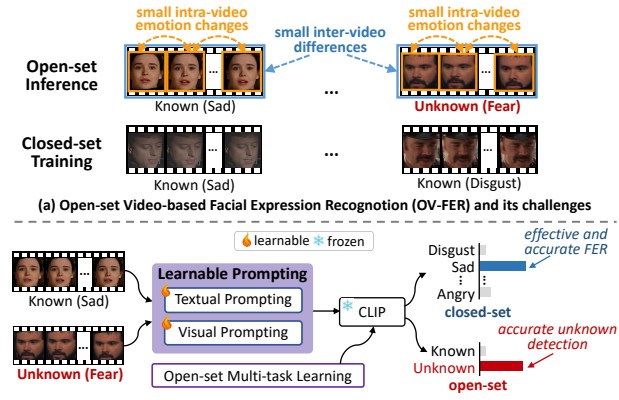

**Figure 1: The motivation and intuitive results of our HESP for OV-FER. Affected by challenges in OV-FER, current methods struggle to capture effective expressive information due to subtle inter-video and intra emotion changes and differences. To address this limitation, HESP consists of a novel learnable textual prompting module, visual prompting module, and an open-set multi-task learning scheme, aiming to augment CLIP to obtain more expression-sensitive representations for both known and unknown emotions in OV-FER.**

critical issue arises: CLIP was originally designed for image-text associations, and its support for video analysis, particularly in the context of human emotion videos, is not addressed appropriately in the vanilla model. To tackle this, a few studies [37, 39] attempt to capture spatial-temporal dynamic information in videos to extend the capability of CLIP on open-set video understanding tasks. For instance, AutoLabel [39] introduces an additional self-attention temporal pooling for CLIP to aggregate frame-level features as video-level features. Open-VCLIP [37] proposes a temporal attention view of each self-attention layer in CLIP to facilitate the aggregation of global temporal information. Despite promising progress, these CLIP video-based methods still achieve unsatisfactory results on OV-FER, where changes in facial expressions both between and within videos can be very subtle [25], as shown in Fig. 1.

To address the above problems when applying the CLIP on OV-FER, we propose a novel human expression-sensitive prompting (HESP) mechanism to significantly upgrade the CLIP by equipping it with the ability to understand facial details and model subtle human expressions effectively, therefore enabling the CLIP-based model to properly deal with unseen emotion categories in the OV-FER task. In general, our contributions are summarized as follows:

- We are the first research to explore the OV-FER task. To tackle the task, we propose a novel Human Expression-sensitive Prompting (HESP) framework comprising textual and visual prompting modules, along with a multi-task learning component. HESP significantly augments CLIP's efficacy in achieving effective OV-FER.
- Methodologically, first, we introduce learnable text prompts to better depict emotions in both known and unknown classes. We also create learnable visual prompts to guide CLIP in focusing on expression-sensitive areas in video frames,

capturing temporal visual expression-sensitive information for both known and unknown data. Then, an open-set multi-task learning scheme is proposed to facilitate the cross-modal prompt interaction, thus further enhancing the HESP's performance in capturing inter-video emotion differences for both known and unknown categories.

- Experimentally, inspired by previous open-set tasks, we define experimental settings of OV-FER, and conduct extensive experiments on 4 open-set OV-FER settings, showing that our approach achieves new state-of-the-art results. Specifically, our approach demonstrates average relative gains of 20.85% in AUROC and 142.81% in OSCR, compared to the baseline method across all 4 OV-FER tasks, confirming the effectiveness and generality of our proposed model.

## 2 Related work

**Video-based Facial Expression Recognition (V-FER).** Compared to static image FER, V-FER extracts dynamic changes of facial expressions and recognizes emotion categories. Currently, many studies use structures like LSTM [19, 22], 3D-CNN [24, 26, 42], Transformer [25, 43] to extract spatio-temporal expression information for V-FER. EST [25] proposes to model the subtle motion of facial expressions in videos by decomposing it into a series of expression snippets. Former-DFER [43] proposes a dynamic Transformer that learns expression features from both temporal and spatial perspectives. However, these methods are limited to closed-set datasets with fixed emotion categories, struggling to be applied in real-world open-set environments with unknown emotions.

**Open-set Recognition.** Current open-set recognition methods are classified as: discriminative models and generative models. Discriminative models adds rejection rules to identify unknown classes [4, 7, 16]. Generative models generate fake data that simulates novel classes during training [20, 27]. Although encouraging results have been achieved, both types of methods show limited generalization abilities to FER task due to the subtle inter-class differences in facial expressions. Recently, there have been studies [41, 46] addressing the issue of open-set image FER. For example, VBExT [46] designs a Bi-Attention Expression Transformer to extract more compact features. Open-set FER [41] treats it as a noisy label detection problem and utilizes attention map consistency to separate open-set samples. However, these methods are not designed to handle videos, which limits their applicability in OV-FER.

**Large-scale Vision-language Models and Prompting Learning.** Pre-trained large-scale vision-language models have become significant in computer vision (CV). These models, for instance CLIP [29] and GPT-4 [1], learning rich visual and linguistic features through pre-training on large-scale image and text data, achieving significant performance improvements in various downstream tasks, such as image classification [45], object detection [33], as well as open-set recognition [21, 30, 34]. Prompt learning (PL) aids large-scale models in adapting to various downstream tasks [31, 38, 45]. CoOp [45] employs learnable text prompts, avoiding manual prompt tuning. VPT [17] uses trainable parameters as vector sequence visual prompts to improve the performance. Despite the progress, existing PL methods are mainly designed for generic image recognition, lacking effective PL for more challenging FER tasks.

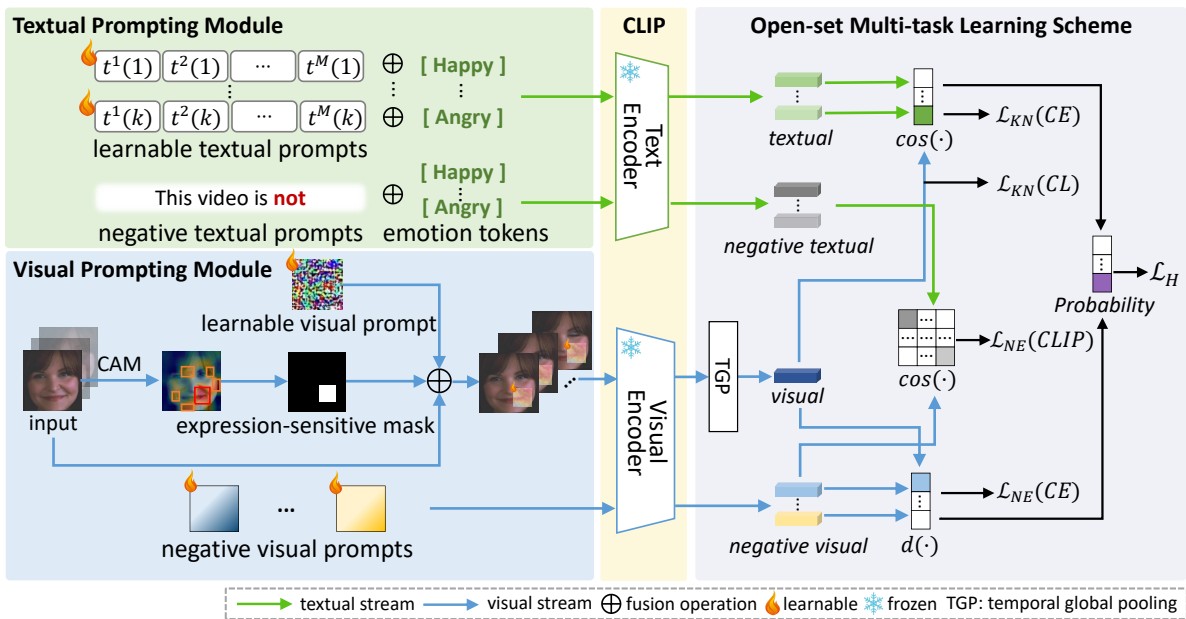

**Figure 2: The pipeline of HESP for OV-FER. HESP first combines textual and visual prompting modules to enhance CLIP for modelling facial expression-sensitive information. Then, an open-set multi-task learning scheme is devised to facilitate interactions between these modules, improving OV-FER performance by exploring both known and unknown emotion cues.**

## 3 Method

### 3.1 Preliminary

***Problem Definition.*** We first define the problem of the OV-FER task. In OV-FER, the training set is denoted as $D_{\text{train}} = \{(V_i, y_i)\}$, where $i$ indexes over samples, $V_i$ represents the $i$-th video sample, $y_i \in Y_{known} = \{1, \ldots, K\}$ is the label of $V_i$, and $K$ is the number of known classes. Following existing open-set settings [7, 46], there are novel, unknown expression categories in the test set. We consider all novel, unknown categories as $Y_{unknown} = \{K + 1\}$. Therefore, the test set is defined as $D_{test} = \{(V_j, y_j)\}$, where $y_j \in Y_{known} \cup Y_{unknown}$, and $j$ indexes over test samples. This task then aims to accurately recognize the category of video $V_j$ from both $Y_{known}$ and $Y_{unknown}$.

***Negative Representations for Open-set Problems.*** Negative representations, opposite to known class representations, are particularly beneficial for discovering unknown patterns in open-set datasets [7, 8, 14]. In open-set data, negative representations help distinguish irrelevant patterns, like background features, enabling better recognition of meaningful, unknown patterns. Therefore, data with patterns can be collected from negative representations of all categories to discover unknown categories. For more details, we refer readers to [7]. In the OV-FER task, since the differences between videos could be very small, it is important to identify the patterns related to negative representations, thereby driving us to follow such a mechanism for implementing the CLIP-based OV-FER approach.

***CLIP Model.*** The vanilla CLIP [29] first uses a Transformer-based text encoder [32], denoted as $\phi_T$, and uses a ViT [13] visual encoder, denoted as $\phi_V$, to extract high-level features from text and

vision inputs, respectively. Suppose we have a textual input $T$ and a visual input $V_i$, the CLIP extract features:

$$F_T = \phi_T(T), \; F_{V,i} = \phi_V(V_i), \tag{1}$$

where $F_T$ and $F_{V,i}$ are extracted textual and visual features, respectively. Then, CLIP correlates the text features with the visual features based on the relationships between the input texts and images. The correlation between $F_T$ and $F_{V,i}$ is estimated by $corr(F_T, F_{V,i})$ where $corr$ is a correlation estimation process. Normally, $corr$ is implemented by inner-product followed by a softmax, and the obtained scores can be used for various zero-shot purposes, like open-set tasks. For example, if the correlation score between an image and the phrase "a photo of a happy expression" is very high, we can consider that this image contains a happy face, even without training or fine-tuning.

### 3.2 Overview of HESP

Fig. 2 illustrates the overview of how our proposed HESP method augments CLIP for the OV-FER task. Specifically, our framework comprises three modules: a novel textual prompting module to enhance the CLIP's text encoder, a novel visual prompting module to improve the CLIP's visual encoder, and an open-set multi-task learning scheme to facilitate learning through the interaction between the textual and visual prompting modules. The textual and visual prompting modules enable CLIP to understand emotion-related descriptions and video data, and the learning scheme enables our approach to perform OV-FER effectively. In the CLIP process, we mainly recast the Eq. (1) by introducing inputs enhanced by textual and visual prompting:

$$F'_T = \phi_T(T_H), \; F'_{V,i} = \phi_V(V_{H,i}), \tag{2}$$

where $T_H$ and $V_{H,i}$ are enhanced inputs based on the textual and visual prompting of our HESP, respectively, and $F'_T$ and $F'_{V,i}$ are related refined high-level features for better OV-FER.

## 3.3 Textual Prompting

In the textual prompting of HESP, rather than re-training the entire text encoder of the CLIP which would be extremely costly, we attempt to introduce learnable features more sensitive for both known and unknown categories in OV-FER.

Formally, to obtain an enhanced text input $T_H$ that could be sensitive to emotion phrases, we introduce learnable representations $\delta_T$ to complement the $T$. As a result, the $T_H$ is defined as:

$$T_H = T \oplus \delta_T, \tag{3}$$

where $\oplus$ represents a fusion operation. The detailed implementations are discussed as follows.

***Textual Prompting for Closed-set Data.*** For closed-set data, regarding the text description $T$ to be associated with visual input, we introduce specific emotional tokens denoted as [CLASS-k] for depicting the $k$-th facial expression. The [CLASS-k] is actually an emotional phrase, such as "happy". Normal CLIP usage applies a template of "a photo of X" where "X" here can be an emotional token [CLASS-k], but it cannot appropriately describe a video with human expressions. As a result, we instead introduce learnable templates to instantiate $\delta_T$.

Using the emotional token $T$ for the closed-set data, we define that $\delta_T$ is a sequence of learnable text representations, *i.e.*,

$$\delta_T(k) = [t^1(k), t^2(k), \ldots, t^M(k)], \tag{4}$$

where $M$ is a pre-defined length of $\delta_T$ and $k$ refers to the same emotion category with [CLASS-k]. In this study, we set $M$ to 16 empirically. As a result, the Eq. (3) will have a form of:

$$T_H(k) = [t^1(k), t^2(k), \ldots, t^M(k), CLASS - k], \tag{5}$$

which means that the $\oplus$ actually implements a concatenation function that attaches together the $\delta_T(k)$ and the token [CLASS-k]. This design can be partially supported by CoOp [45] which demonstrates that learnable tokens can be integrated with normal tokens to obtain better text prompts regarding specific tasks.

Now, by collecting $T_H(k)$ for all the closed-set categories, we have the final representation $T_H = \{T_H(1), T_H(2), \ldots, T_H(K)\}$, where $K$ is the number of known categories for close-set data. The $T_H$ is then fed into the $\phi_T$ for text encoding with CLIP. It is worth mentioning that we calculate $T_H(k)$ independently, which means that $F'_T = \phi_T(T_H) = \{\phi_T(T_H(1)), \phi_T(T_H(2)), \ldots, \phi_T(T_H(K)))\}$.

***Negative Textual Prompting for Open-set Data.*** To discover unknown classes in open-set data, we follow [7] and introduce negative representations. Specifically, we introduce $K$ different tokens, each of which helps discover patterns that do NOT belong to a specific known emotion category. In practice, we apply $K$ fixed negative textual prompts as the tokens to help explore unknown emotion categories in open-set datasets. The negative textual prompts $\bar{T}(k)$ are usually in the form of "This video is not [CLASS-k]", where the [CLASS-k] enumerates all the closed-set emotional categories. Then, we employ the the CLIP text encoder $\phi_T$ to obtain the negative textual features $\bar{F}_{T'} = \{\phi_T(\bar{T}(1)), \ldots, \phi_T(\bar{T}(K))\}$.

## 3.4 Visual Prompting

For the OV-FER task, we augment CLIP by introducing learnable visual representations with attention to expression-sensitive areas. Besides, we introduce a temporal modeling module for this visual prompting, enabling the understanding of emotional motions in a video. Our visual prompting of HESP also does not require the re-training of the visual encoder of CLIP, thus we can still make the best use of the powerful visual feature extractors used in the vanilla CLIP like ResNet [15] or ViT [13] to encode visual features.

Formally, to introduce a refined visual input $V_{H,i}$ that is sensitive to expression areas on frames of a video, we introduce learnable representations $\delta_V$, and we have:

$$V_{H,i} = V_i \oplus \delta_V, \tag{6}$$

where $\oplus$ is a fusion operation. Although in a modality different from texts, we would like to mention that the benefits of adding learnable representations to visual input can also be partially validated by recent research [2, 9, 10, 36, 40]. In the following, we will discuss in detail how this process, for the first time, can be implemented in the OV-FER task.

***Visual Prompting for Closed-set Data.*** For closed-set data, we define that the $V$ is the video input consisting of a sequence of frames. We use the $q$ to index over each frame of a video, thus $V_i(q)$ refers to the $q$-th frame in $i$-th video. Since we aim to focus on expression-sensitive details on $V_i$, we need to first identify what area on a frame contains the expression-sensitive information, so that a learnable representation can complement the visual input $V_i(q)$ by emphasizing the expression-sensitive information for CLIP. To achieve this, we introduce an expression-sensitive mask $\mathcal{M}_i(q)$ for the $q$-th frame.

We define that the expression-sensitive mask $\mathcal{M}_i(q)$ has the value of 1 for areas on expression-sensitive areas of a frame and the value of 0 for other areas. Using this mask, we implement the Eq. (6) as :

$$V_{H,i} = (1 - \mathcal{M}_i(q)) \cdot V_i(q) + \mathcal{M}_i(q) \cdot \delta_V(q), \tag{7}$$

which means that the fusion $\oplus$ is implemented by the weighted sum of $V_i(q)$ and $\delta_V(q)$ *w.r.t.* $\mathcal{M}_i(q)$. For the learnable representation $\delta_V(q)$, we begin with pixel perturbations on the expression-sensitive areas indicated by $\mathcal{M}_i(q)$ and then optimize the parameters within $\delta_V(q)$ during training.

The $\mathcal{M}_i(q)$ enables the learnable representation $\delta_V(q)$ to complement the raw visual input on expression-sensitive areas without disturbing other areas. To obtain a proper mask $\mathcal{M}_i(q)$, we use the popular class activation mapping (CAM) [44] mechanism. More specifically, we use the visual encoder of CLIP to extract CAM and use the results to obtain expression-sensitive areas. Based on the CAM result, we find a rectangular area with a higher score and set the values of the mask inside this area to 1, keeping other values at 0. This rectangular area is defined to have a length of $l$, and $l$ is set to 56. In practice, we surprisingly found that keeping this rectangular area stable throughout the video is more beneficial, thus we only rely on the mask generated in the first frame. Please refer to the supplementary material for more details.

For a video, we extract visual features from each of the frame and aggregate them together to obtain the final refined high-level visual

features. That is, we have: $F'_{V,i} = g(\phi_V(V_{H,i})) = g(\{\phi_V(V_{H,i}(1)), \phi_V(V_{H,i}(2)), \ldots, \phi_V(V_{H,i}(N))\})$, where $N$ refers to the frame amount, and $g(\cdot)$ is a temporal global pooling to aggregate all the results from frame 1 to frame $N$ to obtain global temporal information.

***Negative Visual Prompting for Open-set Data***. For open-set data, we still introduce negative representations. Instead of textual prompting where we have an explicit negation phrase "not", we attempt to apply implicit negative visual representations $\bar{V}_{i,k}$ that need to be optimized together with our HESP promoting modules. Specifically, given $K$ known categories, we start with randomly initialized $K$ tensors to represent the negative representations for each $K$ known category. To ensure consistency with closed-set data inputs for CLIP, we set the dimensions of each implicit negative visual representation as the same with a frame $V_i(q)$, namely $3 \times 224 \times 224$. Then, using the encoder $\phi_V$ in CLIP, we extract the negative visual prompting features as $\bar{F}'_V = \{\phi_V(\bar{V}_{i,1}), \ldots, \phi_V(\bar{V}_{i,K})\}$. After training, although the negative text representations already facilitate the identification of irrelevant visual patterns, the performance could be affected by visual noises. Therefore, further calculating the distances between open-set video input and learned implicit negative visual representations allows us to better identify irrelevant visual patterns and help discover more appropriate unknown patterns.

## 3.5 Optimization: Open-set Multi-task Learning

To fine-tune and optimize HESP, we introduce an open-set multi-task learning scheme consisting of three major objectives: loss for known category learning $\mathcal{L}_{KN}$, loss for negative representation learning $\mathcal{L}_{NE}$, and loss for final prediction $\mathcal{L}_H$, thereby we have the overall learning objective $\mathcal{L}$ as follows:

$$\mathcal{L} = \mathcal{L}_{KN} + \mathcal{L}_{NE} + \mathcal{L}_H. \tag{8}$$

For known category learning, with the visual features $F'_{V,i}$ and textual features $F'_T$ augmented by HESP, we simply follow CLIP's procedure [29] to obtain prediction results as,

$$P_{KN,i} = softmax(cos(F'_{V,i}, F'_T)), \tag{9}$$

where $P_{KN,i}$ is the known class prediction for the $i$-th video, $softmax$ is the softmax operation and $cos$ is cosine similarity. Then, we use the cross-entropy loss to make $P_{KN,i}$ predict known classes well. Besides, we also introduce a supervised contrastive loss to further enlarge inter-class feature distances and reduce intra-class feature distances, which is advantageous for feature learning in open-set data. As a result, the $\mathcal{L}_{KN}$ is written as:

$$\mathcal{L}_{KN} = \sum_i CE(P_{KN,i}, y_i) + CL(F'_{V,i}), \tag{10}$$

where $y_i$ is the known class label for the $i$-th video, $CE$ represents cross-entropy loss, and $CL$ represents a contrastive learning loss. Note that the $CL$ contrasts $F'_{V,i}$ by making features of different known classes away from each other as much as possible.

For negative representation learning, as mentioned previously, we follow [7] and use negative representations to de-emphasize potentially irrelevant patterns for better open-set recognition. Therefore, we will have another prediction, termed $P_{NE,i}$, based on negative representations. We would like to clarify that this prediction relies on the *double negation* concept, that is, we attempt to find data

that is not negative representations. We achieve this by calculating distances between inputs and negative representations:

$$P_{NE,i} = softmax(d(F'_{V,i}, \bar{F}'_V)), \tag{11}$$

where $d$ is Euclidean distance, and $\bar{F}'_V$ is the collection of visual features extracted from learned negative visual representations. Note that $P_{NE,i}$ is also a $K$-classification problem. We also use cross-entropy loss to optimize the prediction $P_{NE,i}$, which encourages the negative representations to encode features of little interest to known class data as well as unknown class data. Additionally, we introduce a contrast CLIP loss [29], represented as $CLIP$, to further align the negative textual features $\bar{F}'_T$ and negative visual features $\bar{F}'_V$ as closely as possible. Then, we have:

$$\mathcal{L}_{NE} = \sum_i CE(P_{NE,i}, y_i) + CLIP(\bar{F}'_V, \bar{F}'_T), \tag{12}$$

It is worth mentioning that the supervised $CL$ loss is not useful here since optimizing the above loss already makes negative representations away from known class features and there is no point in pulling irrelevant features together.

Lastly, we have an overall class prediction $P_{H,i}$:

$$P_{H,i} = (P_{KN,i} + P_{NE,i})/2. \tag{13}$$

We optimize the $P_{H,i}$ using one another cross-entropy loss:

$$\mathcal{L}_H = \sum_i CE(P_{H,i}, y_i). \tag{14}$$

## 3.6 Inference

For inference, we followed existing open-set evaluation strategies [7, 28]. Given a novel video $V_j$ as input, we first follow the HESP and CLIP to extract the visual features $F'_V$, textual features $F'_T$, negative visual features $\bar{F}'_V$, and negative textual features $\bar{F}'_T$. Then, we employ Eq. (9), Eq. (11), and Eq. (13) to calculate the overall class prediction $P_{H,j}$. Now, still following negative-representation methods [7] for open-set problem, we analyze the probability distributions $P_{H,j}$ on both closed-set data and open-set data and use a dynamic thresholding method to distinguish the unknown class from known classes by introducing a calibration-free measure. We refer more details to [7, 28].

# 4 Experiments and analysis

## 4.1 Experiment Setup

*4.1.1 **Datasets**.* For evaluation, we used two challenging video-based FER datasets, *i.e.*, AFEW [12] and MAFW [23].
**AFEW [12]:** The AFEW dataset contains videos from movies and TV series with spontaneous expressions under various conditions. With seven basic emotion labels, *i.e.*, anger, disgust, fear, happiness, sadness, surprise, and neutral, AFEW comprises Train (738 videos), Val (352 videos), and Test (653 videos) splits.
**MAFW [23]:** MAFW is the first large, multimodal, multi-label emotion database containing 11 single emotion categories, *i.e.*, anger, disgust, fear, happiness, neutral, sadness, surprise, contempt, anxiety, helplessness, and disappointment, 32 compound emotion categories and descriptive texts. MAFW collects 10,045 videos from movies, TV dramas and short videos in social medias.

**Table 1: OV-FER results based on 7 basic emotions under different openness.**

| Method | AUROC | | | | | OSCR | | | | |
|---|---|---|---|---|---|---|---|---|---|---|
| | O(5:2) | O(4:3) | O(3:4) | O(2:5) | Mean | O(5:2) | O(4:3) | O(3:4) | O(2:5) | Mean |
| ARPL [7] | 54.49 | 53.67 | 54.60 | 53.03 | 53.95 | 13.10 | 12.62 | 20.27 | 27.96 | 18.49 |
| CSSR [16] | 54.36 | 52.00 | 54.86 | 51.56 | 53.20 | 13.25 | 17.83 | 23.13 | 29.85 | 21.02 |
| DIAS [27] | 51.63 | 51.45 | 51.28 | 52.56 | 51.73 | 15.20 | 16.98 | 22.07 | 31.90 | 21.54 |
| DEAR [3] | 52.86 | 50.84 | 50.60 | 51.33 | 51.41 | 12.25 | 13.85 | 18.19 | 28.07 | 18.09 |
| Open-set FER [41] | 56.90 | 55.05 | 54.56 | 54.47 | 55.25 | 13.47 | 17.10 | 20.13 | 30.30 | 20.25 |
| Open-VCLIP [37] | 58.36 | 61.71 | 60.29 | 58.36 | 59.68 | 26.68 | 33.05 | 36.24 | 41.43 | 34.35 |
| Baseline 1 (only CLIP) | 52.25 | 50.23 | 52.47 | 51.31 | 51.57 | 14.51 | 16.22 | 21.81 | 28.75 | 20.32 |
| **Baseline 1+HESP (ours)** | 54.65 | 59.11 | 57.00 | 60.61 | 57.84 | 22.08 | 32.46 | 36.25 | 46.91 | 34.43 |
| Baseline 2 (CLIP+ARPL) | 55.68 | 53.87 | 55.00 | 53.83 | 54.60 | 14.77 | 16.71 | 21.05 | 28.44 | 20.24 |
| **Baseline 2+HESP (ours)** | 65.18 | 65.42 | 62.67 | 64.27 | 64.39 | 29.48 | 39.23 | 44.80 | 53.40 | 41.73 |

**Table 2: OV-FER results based on 11 emotions under various openness**

| Method | AUROC | | | | | OSCR | | | | |
|---|---|---|---|---|---|---|---|---|---|---|
| | O(8:3) | O(6:5) | O(5:6) | O(3:8) | Mean | O(8:3) | O(6:5) | O(5:6) | O(3:8) | Mean |
| ARPL [7] | 56.58 | 52.21 | 49.06 | 53.47 | 52.83 | 17.44 | 26.97 | 25.89 | 30.54 | 25.21 |
| CSSR [16] | 57.51 | 53.07 | 53.17 | 54.74 | 54.62 | 26.37 | 26.98 | 23.85 | 31.58 | 27.20 |
| DIAS [27] | 52.39 | 50.18 | 53.39 | 50.07 | 51.51 | 19.22 | 23.58 | 23.09 | 33.26 | 24.79 |
| DEAR [3] | 42.82 | 45.11 | 37.72 | 44.69 | 42.59 | 14.42 | 20.20 | 17.65 | 28.71 | 20.25 |
| Open-set FER [41] | 54.16 | 52.71 | 52.67 | 52.57 | 53.03 | 15.40 | 21.31 | 18.59 | 29.29 | 21.15 |
| Open-VCLIP [37] | 58.70 | 60.01 | 63.40 | 58.59 | 60.18 | 30.87 | 36.02 | 37.41 | 43.60 | 36.98 |
| Baseline 1 (only CLIP) | 53.53 | 52.29 | 56.93 | 52.07 | 53.71 | 16.50 | 21.14 | 20.99 | 27.93 | 21.64 |
| **Baseline 1+HESP (ours)** | 61.23 | 56.93 | 62.87 | 56.69 | 59.43 | 30.52 | 35.52 | 38.17 | 43.77 | 37.00 |
| Baseline 2 (CLIP+ARPL) | 54.03 | 52.08 | 56.05 | 53.03 | 53.80 | 16.63 | 21.15 | 18.75 | 30.28 | 21.70 |
| **Baseline 2+HESP (ours)** | 65.97 | 61.63 | 66.78 | 64.88 | 64.82 | 41.05 | 40.29 | 47.84 | 52.80 | 45.50 |

*4.1.2* ***OV-FER Task Settings****.* Following the existing open-set settings [28, 46], we divide the expression categories in AFEW and MAFW into known and unknown classes and design four different OV-FER tasks. For better evaluation, we introduce the concept of openness [28, 46] as: $O(K : U) = 1 - \sqrt{K/(K+U)}$. $K$ and $U$ are the number of divided known and unknown expression classes, respectively. A larger $O$ indicates more unknown emotions in open-set data. The four OV-FER tasks are as follows:

**1) OV-FER with 7 basic emotions.** In AFEW, we divide the known classes and unknown class sets, according to four different openness values, *i.e.* , $O(5 : 2) = 0.15$, $O(4 : 3) = 0.24$, $O(3 : 4) = 0.35$, $O(2 : 5) = 0.47$.

**2) OV-FER with 11 emotions.** In MAFW, known and unknown classes are segregated using four distinct openness values: $O(8 : 3) = 0.15$, $O(6 : 5) = 0.26$, $O(5 : 6) = 0.33$, $O(3 : 8) = 0.48$.

**3) OV-FER with fusion datasets.** To simulate more challenging open-set scenario, we combined AFEW with MAFW, using 7 basic emotions as known classes and the other 4 emotions as unknown classes, with the openness $O(7 : 5) = 0.24$.

**4) OV-FER with compound emotions.** For more challenging settings, we used the 7 basic emotions in MAFW as known classes and 9 compound emotions in MAFW as unknown classes, with the openness $O(7 : 9) = 0.34$.

It is worth noting that: 1) under each task, all unknown emotion classes are removed from the training set; 2) we randomly divide

known and unknown classes 5 times according to each openness and report the average results for each openness.

*4.1.3* ***Evaluation Protocols****.* Following the existing work [7, 46], we selected the Area Under the Receiver Operating Characteristic Curve (AUROC) and the Open-Set Classification Rate (OSCR) as our evaluation metrics.

*4.1.4* ***Implementation Details****.* The model was implemented on Ubuntu 20.04 using the PyTorch framework on the NVIDIA GeForce RTX 4090. Facial videos underwent preprocessing with face detection and alignment [11], followed by cropping to 224x224 size. Training utilized 200 epochs with stochastic gradient descent (SGD) optimizer with momentum. The learning rate initially set to 0.01, decreased by a factor of 0.1 every 30 epochs during training. For a fair comparison, we followed exsiting open-set inference testing method [7, 28].

## 4.2 Overall Performance

To comprehensively evaluate the performance of the proposed HESP for OV-FER, we compared it with several open-set image recognition techniques including ARPL [7], CSSR [16], DIAS [27], and Open-set FER [41], as well as open-set video recognition methods including DEAR [3] and Open-VCLIP [37]. We also considered the integrated method CLIP alone and CLIP+ARPL as baselines for comparison. Extensive experimental results confirm the versatility

**Table 3: OV-FER results of the fusion datasets. Openness O(7:5)=0.24.**

| Method | AUROC | OSCR |
|---|---|---|
| ARPL [7] | 56.10 | 15.31 |
| CSSR [16] | 51.29 | 17.56 |
| DIAS [27] | 56.13 | 15.56 |
| DEAR [3] | 55.14 | 15.19 |
| Open-set FER [41] | 57.88 | 13.03 |
| Open-VCLIP [37] | 59.28 | 30.17 |
| Baseline 1 (only CLIP) | 59.25 | 14.28 |
| **Baseline 1+HESP (ours)** | 62.44 | 33.39 |
| Baseline 2 (CLIP+ARPL) | 57.45 | 15.02 |
| **Baseline 2+HESP (ours)** | **68.08** | **43.68** |

**Table 4: OV-FER results of compound emotions. Openness O(7:9)=0.34.**

| Method | AUROC | OSCR |
|---|---|---|
| ARPL [7] | 55.37 | 15.85 |
| CSSR [16] | 60.00 | 17.64 |
| DIAS [27] | 51.63 | 15.99 |
| DEAR [3] | 40.44 | 13.18 |
| Open-set FER [41] | 53.70 | 14.81 |
| Open-VCLIP [37] | 60.98 | 33.32 |
| Baseline 1 (only CLIP) | 57.10 | 16.09 |
| **Baseline 1+HESP (ours)** | 67.36 | 37.94 |
| Baseline 2 (CLIP+ARPL) | 57.33 | 16.41 |
| **Baseline 2+HESP (ours)** | **72.42** | **47.24** |

of our HESP method, enhancing the performance of various CLIP frameworks in OV-FER. Results across four OV-FER tasks are presented in Tables 1-4, with the best (in **Bold**) and second-best results (in Underlined) highlighted.

*4.2.1* ***Experiments on 7 Basic Emotions in AFEW***. Our proposed framework consistently outperforms existing open-set recognition methods across all openness settings, as shown in Table 1. Compared with two baselines, our method significantly improves both AUROC and OSCR, especially on Baseline 2 (CLIP+ARPL), where the OSCR is improved by TWO times, showcasing the effectiveness of our novel HESP in extending various CLIP frameworks for OV-FER. We noted a correlation between increasing openness (*O*) and rising OSCR, indicating improved recognition accuracy of known classes as class numbers decrease.

*4.2.2* ***Experiments on 11 Emotions in MAFW***. Table 2 shows the overall performance of our method and other state-of-the-art open-set methods on 11 emotions in MAFW under four various openness. Despite handling more emotional categories and challenging wild videos, our method outperforms the second best method, Open-VCLIP [37], with relative improvements of 7.71% and 23.04% on AUROC and OSCR, respectively. Optimal performance was observed at openness ($O(5:6) = 0.33$), highlighting the ability of our method to more challenging open-set environments.

*4.2.3* ***Experiments on Fusion Datasets.*** To validate our method's effectiveness in broader open-set scenarios, experiments on fusion datasets were conducted, with results in Table 3. Compared to Open-set FER [41], our method achieves significant improvement, with a relative improvement of 17.62% and 235.23% in AUROC and OSCR, showcasing its ability to process video information, learn dynamic facial expressions, and handle diverse open situations robustly.

*4.2.4* ***Experiments on Compound Emotions.*** We evaluated the performance on more challenging compound emotions, as shown in Table 4. Our method, when added to Baseline 1 (only CLIP), surpasses it relatively by 17.97% and 135.80% in AUROC and OSCR. Integrating it into Baseline 2 (CLIP+ARPL) relatively boosts AUROC and OSCR by 26.32% and 187.87%, respectively. Moreover, most methods displayed low OSCR metrics, indicating subpar performance in closed-set compound emotion recognition, except for our HESP, showcasing significant generalization.

## 4.3 Ablation Studies

*4.3.1* ***Effects of Different Modules.*** In Table 5, we gradually added different components to the baseline on the 7 basic emotion OV-FER task. The Textual Prompting Module (TP) improved AUROC and OSCR by 5.68% and 74.46%, relatively, by providing beneficial text information. Further adding the Visual Prompting Module (VP) further relatively improved AUROC and OSCR by 11.59% and 18.18%, respectively. Both modules emphasize capturing expression-sensitive information for both closed-set and open-set data.

**Table 5: Effects of difference modules in HESP on OV-FER.**

| Baseline 2 (CLIP+ARPL) | TP | VP | AUROC | OSCR |
|---|---|---|---|---|
| √ | | | 54.60 | 20.24 |
| √ | √ | | 57.70 | 35.31 |
| √ | √ | √ | **64.39** | **41.73** |

*4.3.2* ***Effects of Different Losses in the Open-set Multi-Task Learning.*** In Table 6, we analyzed the contributions of different losses on the 7 basic emotion OV-FER task. As shown in Eq. (10) and Eq. (12), both $\mathcal{L}_{KN}$ and $\mathcal{L}_{NE}$ consist of two types of losses, and we conduct experiments sequentially. Analyze this table, using only $\mathcal{L}_{NE}$ increased AUROC and OSCR by 8.63% and 37.86% respectively compared to using only $\mathcal{L}_{KN}$, demonstrating the effectiveness of exploring open-set information from negative representations. Finally, the best result was achieved by fusing all losses through $\mathcal{L}_H$.

*4.3.3* ***Effects of Different Settings in Textual and Visual Prompting Modules.*** Fig. 3 reports experimental comparison of different textual prompting and visual prompting methods. In Fig. 3(a), fixed templates has better performance than learnable representations in the negative textual prompting, validating the effectiveness of introducing intuitive negative class information in open-set scenarios. In addition, to verify the effectiveness of our proposed learnable visual representations for closed-set data, we compared it with two other visual prompt representations, *i.e.*, padding around each frame and

Table 6: Effects of different losses in HESP on OV-FER

| $\mathcal{L}_{KN}$ | | $\mathcal{L}_{NE}$ | | $\mathcal{L}_H$ | AUROC | OSCR |
|---|---|---|---|---|---|---|
| CE | CL | CE | CLIP | | | |
| √ | | | | | 54.99 | 24.40 |
| √ | √ | | | | 55.60 | 25.01 |
| | | √ | | | 60.03 | 33.66 |
| | | √ | √ | | 60.40 | 34.48 |
| √ | √ | √ | √ | √ | **64.39** | **41.73** |

random patches within videos. The results are shown in Fig. 3(b), demonstrating that our learnable visual representations performs the best. We also conducted ablation experiments with various sizes of our learnable visual prompts to 14×14, 28×28, 56×56 and 112 ×112, respectively, as shown in Fig. 3(c). Obviously, the best result was achieved when the size was 56×56.

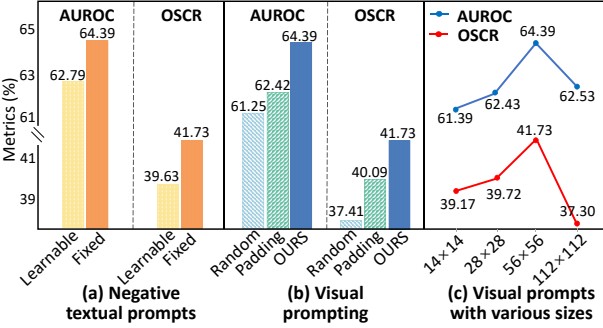

Figure 3: Various settings in textual and visual prompting.

### 4.3.4 *Experiments on Large Dataset and Inference Speed.*
To further validate the effectiveness and universality of our method, we evaluated our HESP on a large dataset DFEW [18] containing 7 basic emotions, and the results are shown in Table 7. Our method still performs best, relatively surpassing Open-VCLIP [37] 17.63% and 36.30% on AUROC and OSCR, respectively. Meanwhile, we compared the full model inference speed of various methods in Table 7. Analysis of the table reveals that our method significantly improves model performance while reducing time consumption.

Table 7: OV-FER evaluation on the DFEW dataset.

| Method | Performance | | Computation cost |
|---|---|---|---|
| | AUROC | OSCR | Inference speed (s) |
| ARPL [7] | 54.46 | 29.12 | 0.021 |
| Open-set FER [41] | 54.78 | 27.78 | 0.077 |
| Open-VCLIP [37] | 56.83 | 39.92 | 0.131 |
| **Baseline 2+HESP (ours)** | **66.85** | **54.41** | 0.028 |

## 4.4 Visualization

### 4.4.1 *Visualization on Different Probability Scores.* Fig. 4
shows the probability scores for known and unknown emotion detection on open-set data through ARPL [7], CLIP+ARPL, Open-VCLIP [37], and our HESP. We normalized the probability scores to

[0,1] for comparison. Compared to other methods where the probability scores of known and unknown samples highly overlap, our HESP effectively separated their probabilities. This demonstrates that our HESP learns discriminative facial expression information and successfully introduces open-set information through negative category cues, thus effectively discovering unknown emotion categories in open-set data.

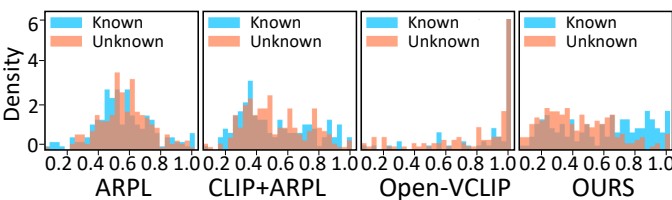

Figure 4: Known and unknown probability distributions.

### 4.4.2 *Visualization on Various Features.* Fig. 5 illustrates high-level facial expression features extracted by ARPL [7], CLIP+ARPL, Open-VCLIP [37], and our HESP on the 7 basic emotion OV-FER task under the openness $O(3:4)$. Obviously, compared to others, the features extracted by our HESP show clear separation across all emotion classes, proving its capability to extract more discriminated, expression-sensitive features, effectively distinguishing known and unknown classes, and accurately identifying known classes.

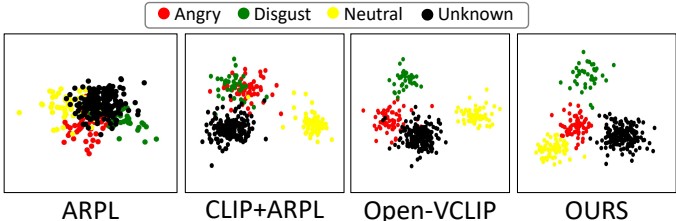

Figure 5: Visualization of facial expression features extracted by different methods. Owing to space limitation, we only present the results on the 7 basic emotion OV-FER task under the openness O (3:4). More visualization results can be shown in the supplementary material.

## 5 Conclusion

In this paper, we propose a Human Expression-Sensitive Prompting (HESP) mechanism to enhance the model's understanding of subtle differences and nuanced human expression patterns in video sequences. HESP comprises a textual prompting module and a visual prompting module, both designed to capture emotion-related patterns within videos and supplement CLIP's original input to better characterize known or unknown emotions. We also devise an open-set multi-task learning scheme to facilitate interactions between the textual and visual prompting modules, augmenting the understanding of novel human emotion-related patterns in video sequences. Extensive experiments on four OV-FER tasks demonstrate that HESP significantly outperforms other state-of-the-art open-set recognition methods. In future work, we will extend HESP to explore multimodal open-set recognition as well as more generalized category discovery in unknown multi-class recognition.

## Acknowledgments

This work was supported by the National Natural Science Foundation of China grant (62076227), Natural Science Foundation of Hubei Province grant (2023AFB572) and Hubei Key Laboratory of Intelligent Geo-Information Processing (KLIGIP-2022-B10).

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
