# OpenReview forum: "Open-Set Video-based Facial Expression Recognition with Human Expression-sensitive Prompting"
_acmmm.org/ACMMM/2024/Conference — MM2024 Poster_

### Official Review · Reviewer_99CH · 2024-05-11

**Rating:** 3
**Confidence:** 3

**Summary:**

The paper focuses on an open-set video-based facial expression recognition task. The categorical classes of facial expressions that are not included in the training set are considered the unknown classes. It aims to distinguish the unknown categorical facial expressions from the test set. The authors proposed a new method called Human Expression-Sensitive Prompting (HESP), which integrates textual and visual modules to improve the ability of the CLIP model to process and recognize both known and unknown facial expressions. Textual prompting and visual prompting are employed to upgrade the performance. A multi-task learning scheme is employed to achieve the training.

**Strengths:**

The authors introduce a new topic into video-based facial expression recognition (V-FER) tasks. The proposed method seems effective.

**Limitations:**

Though the performance upgrade seems obvious, the innovation to upgrade the existing CLIP method is incremental and trivial. The following are some suggestions to improve this paper.

1. It will be better to first write down the problem that the video-based facial expression recognition faces to introduce the specific design of your proposed modules.

2. Line 144 ~ Line 172: It is cumbersome to summary the novelty in such long paragraphs. Please simplify the proposed novelty. In addition, the difficulties and proposed methods can be clearly clarified in the first few paragraphs of the introduction section.

3. In line 201, the authors mentioned that the VBExT and Open-set FER can not handle videos. That is false. They can deal with video-based FER once they replace the backbone networks.

4. Line 273: The authors mentioned that the negative representations help distinguish irrelevant patterns from the relevant ones, enabling better recognition of meaningful, unknown patterns. It infers that the negative representations help to recognize the unknown classes. Therefore, it is unclear why the authors hold the view that the negative representation will be de-emphasized in the OV-FER task, even though the differences between the videos could be very small.

5. Line 317: ,including -> like

6. About Experiments: It is worth providing the benchmark and testing the proposed method in a large-scale dataset with 7 categorical discrete emotions, such as the DFEW dataset.

   Dfew: A large-scale database for recognizing dynamic facial expressions in the wild, in ACM MM 2020.

7. Related work: Please compare the prior works with your proposed method in the section of related work.

8. About the idea: The proposed prompting approach can be easily extended into image-based facial expression recognition. Why does the author emphasize this method is specifically designed for video-based facial expression recognition?

9. In line 376, the authors said the designed method can be partially supported by CoOp. Please clarify this statement. In addition, the proposed method looks like a simple combination of some modules. Can the authors clarify how do they combine organically to present the upgraded performance?

**Suitability:**

2

---

### Official Review · Reviewer_kCLH · 2024-05-29

**Rating:** 5
**Confidence:** 3

**Summary:**

The author studied the open-set facial expression problem and proposed using the HESP mechanism, which utilizes CLIP to enhance performance in addressing the open-set issue.

**Strengths:**

The paper is written clearly and is easy to understand.

This work extends the original image-based open-set FER [40] to a video-based approach.

The use of CLIP significantly improves the performance of open-set facial expression recognition (FER).

**Limitations:**

The overall narrative of this paper is not very convincing to me, which prevents me from giving it a high ranking. The issue of small inter-video differences is crucial to open-set FER, but there is no deep discussion on how CLIP addresses this problem.

Figure 1 of this paper illustrates the small intra-video changes problem. However, this issue is neither mentioned nor addressed in the rest of the paper. If the intention of Figure 1 is similar to that of Figure 1 in [40], I believe there is no need to mention small intra-video changes.

The paper uses a visual prompting module to convert a video input into an image input. However, I am curious whether the introduction of CLIP will also enhance the performance of the original image-based open-set FER. It would be beneficial to include some discussion on this point.

**Suitability:**

2

---

### Official Review · Reviewer_kMHx · 2024-05-30

**Rating:** 4
**Confidence:** 3

**Summary:**

This manuscript develops a human expression-sensitive prompting mechanism to model video-based facial expression details by using CLIP-based OV-FER method.  The experiments have been conducted to validate the effectiveness of the proposed method. However, there are some limitations as follows:
1. In Fig. 2, how to compute the loss between different outputs in detail?
2. In Eq. 8, what are the definitions of different loss functions? How do the authors balance the trade-off between different loss?
3. Please clarify the introduction and abstract clearly.

**Strengths:**

The task and idea are interesting.

**Limitations:**

The authors should polish the manuscript.

**Suitability:**

2

---

### Official Review · Reviewer_6RMW · 2024-06-02

**Rating:** 4
**Confidence:** 2

**Summary:**

To deal with OV-FER, this paper proposes a novel Human Expression-Sensitive Prompting (HESP) mechanism, including a textual prompting module with learnable prompt representations, a visual prompting module that encodes temporal emotional information from video frames, an open-set multi-task learning scheme that boosts interactions between the textual and visual prompting modules.

**Strengths:**

- The paper is largely clearly written and makes an easy read.
- The paper addresses a significant limitation in the field of V-FER.
- The proposed approach is intuitive and solid.

**Limitations:**

- The authors need to evaluate how well the proposed approach scales with larger datasets and a more diverse set of expressions. Is there a point at which the model's performance increases or decreases?
- How well can the model deal with ambiguous or mixed expressions? In real-world scenarios, expressions may be ambiguous or mixed. It is recommended to evaluate how the model handles such cases and whether it can accurately classify these ambiguous expressions or recognize them as unknown.
- While the paper reports significant improvements in AUROC and OSCR, I’m wondering if these metrics are sufficient to capture all aspects of performance relevant to OV-FER tasks. Are there other metrics that should be considered?
- Based on my understanding, human expressions can vary widely based on cultural, individual, and contextual factors. Please analyze whether the proposed approach is sensitive to such variations and how it addresses potential biases in the training data.
- Minor: What’s the latency of running inference with the model compared with existing works? The latency should be low enough for the model deployment for real-world applications.

**Suitability:**

3

---

### Official Review · Reviewer_AmPj · 2024-06-03

**Rating:** 6
**Confidence:** 1

**Summary:**

This paper proposes a novel Human Expression Sensitive Prompt (HESP) mechanism aimed at enhancing the model's understanding and recognition of facial expression details in videos. The main concept behind the HESP is to introduce new modules to enhance the open set recognition ability of CLIP models. Through intensive experiments, the authors show that their new modules and approach results in significant improvement over baseline methods.

**Strengths:**

1). Novelty: The HESP mechanism proposed in the paper is conceptually novel, which is not complex yet effectively enhances the recognition capability for known and unknown emotion categories by combining text and visual prompt modules with an open-set multi-task learning scheme.

2). Significant performance improvement: According to the experimental results, HESP has achieved significant performance improvements on multiple standard datasets, especially the improvements in AUROC and OSCR metrics, which prove the impressive effects of the method.

3). Extensive experimental validation: The authors have conducted extensive experiments on different datasets and different openness settings, which helps to prove the robustness and universality of HESP. The paper also includes an ablation study, which helps readers understand the specific impact of different components and design choices on the final performance.

**Limitations:**

I think the overall idea and empirical studies in this paper have no severe limitations.  But I have some suggestions for paper writing:

1). Figure 2 which demonstrates the whole pipeline seems to be too complex. It is better that the author can simplify the pipeline in some way to make it easier to understand.

**Suitability:**

3

---

### Official Review · Reviewer_rVN7 · 2024-06-04

**Rating:** 5
**Confidence:** 3

**Summary:**

This paper introduces a novel Open-set Video-based Facial Expression Recognition (OV-FER) task, aimed at identifying both known and previously unseen facial expressions in videos. To address this challenge, the authors propose a Human Expression-Sensitive Prompting (HESP) mechanism, which enhances the CLIP model's ability to capture subtle facial expression details. HESP includes a textual prompting module, a visual prompting module, and an open-set multi-task learning scheme. Experimental results demonstrate that HESP significantly improves performance over existing methods across four different OV-FER tasks.

**Strengths:**

The study introduces a novel Human Expression-Sensitive Prompting (HESP) mechanism designed to enhance the model's ability to understand subtle differences and nuanced human expression patterns in video sequences. The study also proposes an open-set multi-task learning scheme that facilitates interactions between the textual and visual prompting modules.

The effectiveness of the proposed HESP mechanism and open-set multi-task learning scheme is demonstrated through extensive experiments on multiple OV-FER. The experimental results show that the proposed approach significantly outperforms other SOTA open-set recognition methods.

**Limitations:**

The proposed HESP mechanism involves a sophisticated combination of textual and visual prompting modules along with a multi-task learning scheme. This complexity can make the implementation and tuning of the model more challenging and time-consuming compared to simpler models.

While the paper demonstrates the effectiveness of HESP on several datasets, it focuses primarily on specific video-based facial expression recognition tasks. The generalizability of the approach to other types of open-set recognition tasks or different domains (e.g., non-facial expressions or different video contexts) remains to be fully explored and validated.

**Suitability:**

3

---

### Meta-Review · Area_Chair_MZsB · 2024-07-01

**Recommendation:** Accept (Poster)
**Confidence:** 5

**Metareview:**

I recommend accept based on the final ratings of all reviewers. Please carefully revise the paper  according to the reviewers' comments.

---

### Meta-Review · Senior_Area_Chairs · 2024-07-10

**Recommendation:** Accept (Poster)
**Confidence:** 4

**Metareview:**

This paper recieved mixed ratings (5 positive and 1 negative) initially. After rebuttal, 5 reviewers tend to accept the paper while one reviewer still questioned novelty of the paper. SAC and AC carefully checked the reviews and rebuttal and recommend acceptance of the paper.